# Site Quality for *Araucaria angustifolia* Plantations with Subtropical Cambisol Is Driven by Soil Organism Assemblage and the Litter and Soil Compartments

Tancredo Souza [1,2,*], Mário Dobner, Jr. [3], Diego Silva Batista [2], Damiana Justino Araujo [2], Gislaine dos Santos Nascimento [4] and Lucas Jónatan Rodrigues da Silva [5]

1 Centre for Functional Ecology, Department of Life Sciences, University of Coimbra, 3000-456 Coimbra, Portugal
2 Postgraduate Program in Agroecology, Department of Agriculture, Federal University of Paraiba, Bananeiras 58220-000, Brazil; diegoesperanca@gmail.com (D.S.B.); damiana.justino@gmail.com (D.J.A.)
3 Department of Agriculcure, Biodiversity, and Forests, Federal University of Santa Catarina, Florianópolis 88580-000, Brazil; dobnerjr@gmail.com
4 Postgraduate Program in Soil and Water Management, Department of Soils, Federal Rural University of Semiarid, Mossoró 59626-590, Brazil; gislaynesantos30@gmail.com
5 Postgraduate Program in Agronomy, Department of Forest, Soil, and Environmental Resources, São Paulo State University, Botucatu 18603-970, Brazil; lucasrodriguesgestorambiental@gmail.com
* Correspondence: tancredo.souza@ufsc.br

**Abstract:** Different site quality levels in *Araucaria angustifolia* (Bert.) O. Kuntze plantations may influence the soil organisms and the interaction between litter and soil chemical properties by providing habitats and nutrients in different pathways. Our aim here was to understand the effect of site quality level in the interaction among litter, soil–solid phase, and organism assemblage on *A. angustifolia*, Campo Belo de Sul, Santa Catarina, Southern Brazil. In the low site quality, the litter deposition, litter K content, litter Ca content, soil organic matter, soil P content, soil K content, and soil exchangeable Ca reduced by 50.50, 49.54, 11.89, 20.51, 11.74, 61.18, and 35.18%, respectively, when compared to the high site quality. Nonmetric multidimensional scaling (NMDS) grouped the influence of site quality degree into three groups, considering the dissimilarities among soil organisms. The ordination of the soil organisms, richness, and Shannon's diversity in each studied site quality degree had a stress value of 0.08. The structural equation models showed that the loss of site quality had a negative relationship with soil organism assemblage and soil and litter compartments. Our study highlights the fact that a fertile soil, a soil enriched in organisms, and enough litter support the forest productivity.

**Keywords:** k-factor; soil ecology in subtropical ecosystem; subtropical Cambisol properties





## 1. Introduction

Site quality is a variable or measure used in forestry to assess the productivity of a forest site, particularly in terms of tree height growth. It represents the average height that dominant trees (in our study, *Araucaria angustifolia* (Bert.) O. Kuntze) achieve under growing conditions, for example, soil ecosystem [1]. Exploring the soil ecosystem involves examining its various compartments—litter, mineral and organic soil fractions (referred to collectively as the soil compartment), organisms, water, and gases—individually to understand the impact of site quality variations on each. It is widely documented that site quality notably impacts both the litter and soil compartments, as well as the assemblage of soil organisms, due to positive plant–soil–litter interactions [2]. The understanding of site quality's role is still limited, largely due to the complex array of soil functions and interactions among soil compartments. Additionally, the lack of comprehensive scientific studies, particularly those utilizing data from long-term field experiments examining the

interplay of organisms, litter, and soil in response to varying site quality, further contributes to this challenge [3]. Soil compartments, such as (i) litter, which may be characterized by litter deposition (kg m$^{-2}$) and k-factor (years$^{-1}$) to assess the input of organic matter and the nutrient cycling process; (ii) soil–solid phase, which may be characterized by the chemical properties that determine soil fertility and the soil capacity to sustain high biomass production; and (iii) soil organism community composition, which may be characterized by an abundance of insects, arachnids, myriapods, etc. that determine the structure of the soil food web and ecosystem services, are among the most important compartments promoting site quality and generating positive interactions between plants, soil, and litter [1].

The ecological significance of these three compartments is attributed to their innate traits (e.g., litter with its decomposability, soil with its fertility, and organisms with their services) to promote habitat and energy supply that enable litter to supply the nutrient cycling process, soil to sustain plant production, and soil organisms to provide better litter trituration, bioturbation, biological control, and nutrient cycling into subtropical soils [4]. For instance, litter deposition has been known to increase nutrient input, thus stimulating soil and plant nutrient bioavailability over the years [5]. Nutrient input creates an energy supply for soil organisms. Thus, they act by transforming and decomposing the litter residue through mechanisms such as physical trituration and enzymatic decomposition. Finally, a robust soil biota community is created in a way similar to the nutrient content hypothesis described by Souza et al. [3].

Despite such positive interactions among litter, soil, and soil organisms, most evidence of the role of site quality is often based on only one of these aimed compartments, gathered from short-term experiments [6,7]. Such evidence has increased our understanding of the role of site quality in processes related to litter dynamics, soil and plant nutrient cycling, k-factor, and soil organism abundance [8]. However, these findings do not consider the potential of site quality (by the lack of studies considering well-defined site quality degrees) as a driving factor for litter quantity and quality, soil chemical properties, and soil organism assemblage, and are far from representing the reality in subtropical ecosystems. In subtropical ecosystems, recent research has advanced our understanding of soil dynamics and site quality [2]. Studies show that soil fertility and productivity are influenced by climate, parent material, vegetation cover, and human activities [3–5]. Soil biodiversity and organisms play a crucial role in nutrient cycling and ecosystem functioning [8]. Subtropical soils also contribute significantly to carbon sequestration and are impacted by land-use changes like deforestation and urbanization [7]. However, long-term studies focusing on plantations of *A. angustifolia* in this context are rare [9], and this study contributes to increasing the knowledge in such ecosystems.

*Araucaria angustifolia* is currently defined as a critically endangered native tree species from southern Brazil through intense timber exploitation during the 20th century [10]. In addition to the historic interest in the productive potential of *A. angustifolia* and the quality of its wood [11], soil fertility characterization, which is one of the main factors contributing to *A. angustifolia* growth and yield, is still little known. Moreover, due to the lack of knowledge about the ecology of *A. angustifolia*, mistakes have been made in selecting sites for establishing plantations of this endangered tree species [3] and in the lack of soil management practices inside these sites to condition the soil ecosystem [12].

Our aim here was to understand the effect of site quality level in the interaction among litter, soil–solid phase, and organism assemblage on *A. angustifolia* cultivated in subtropical Cambisol. To this end, we analyzed the soil chemical properties [13], litter properties [14], and soil organism abundance [15] for three site quality degrees. We hypothesized that the high-quality sites would present high litter deposition with a low lignin content and high macronutrient content (N, P, and K) and soil exchangeable cation content. It resulted in an increased release of available nutrients, as well as a concomitant increase in soil organism richness, and diversity through a positive interaction among the three studied soil compartments (litter, soil, and organisms).

## 2. Materials and Methods

### 2.1. Experimental Design Overview

We conducted a field experiment with established *A. angustifolia* using a completely randomized block design with three site quality levels: (1) low site quality (L-SQ); (2) average site quality (A-SQ, Control); and (3) high site quality (H-SQ) (Table 1). Site quality was estimated in terms of the tree average height: L-SQ: tree height lower than 13 m; A-SQ: tree height ranging from 13.1 to 18.0 m; and H-SQ: tree height higher than 18.1 m [16]. Each site quality level was replicated eleven times using circular plots with 500 m² for two consecutive years. *Araucaria angustifolia* was selected as a model plant because this endangered tree species covered an area of 233,000 km² of Brazilian territory in 1920. Nowadays (January 2023), it has lost 97% of its original area to logging, agricultural purposes, biological invasion, and the spread of *Pinus* plantations in the states of Paraná, Santa Catarina, and Rio Grande do Sul [3]. The mean annual temperature of the experimental sites was +15 °C. We registered a total annual precipitation of 1750 mm (from August 2021 to August 2022), within average historical records. The soil type in the experimental area was classified as subtropical humic Cambisol [17] Köppen's classification defines the climate of the experimental area as a humid subtropical (Cfb) type [18].

**Table 1.** Main description of the studied site quality levels (low, average, and high site qualities) with subtropical Cambisol in Campo Belo do Sul and Capão Alto, Santa Catarina, Brazil.

| Site | Altitude (m) | Slope (%) | Clay Content (g kg⁻¹) | Silt Content (g kg⁻¹) | Sand Content (g kg⁻¹) | Stand Age (Years) | Tree Dominant Height (m) |
|---|---|---|---|---|---|---|---|
| Low quality | 1200 (53) | 12.1 (1.4) | 29.5 (1.7) | 331.6 (9.2) | 638.9 (12.8) | 27.5 (2.8) | 12.7 (1.2) |
| Average quality | 1172 (102) | 11.9 (2.1) | 31.5 (3.2) | 326.6 (5.2) | 641.9 (15.7) | 29.0 (3.2) | 16.0 (2.0) |
| High quality | 1135 (89) | 12.0 (0.9) | 29.6 (2.1) | 339.7 (11.4) | 630.7 (18.2) | 28.8 (2.6) | 20.8 (3.2) |

Standard deviation in parenthesis.

### 2.2. Litter Material and Greenhouse Conditions to Prepare Standard Litter Material

The standard litter material for the litter assay was prepared by considering litter from two native plant species (high C:N ratio—*Mimosa scabrella* and low C:N ratio—*A. angustifolia*). Both plant species were grown in plastic pots containing 4 L of autoclaved sand under greenhouse conditions (25/20 °C day/night temperatures) for 8 weeks. Briefly, seeds of both plant species were surface-sterilized with 0.5% NaOCl (Merck, Oakville, ON, Canada). Leaves were harvested and air dried at 40 °C until a constant dry biomass. The standard litter material was defined as dried leaves free of pathogens and of the same age (8 weeks). We decided to use a standard material to ensure three important aspects: (i) maintaining QA/QC quality by using litter material of the same age for two materials by considering their C/N ratio; (ii) ensuring the litter material was free of pathogens and decomposers by collecting it under aseptic conditions in a greenhouse, thus preventing variations in decomposition stages that could significantly impact the litter decomposition assay; and (iii) maintaining similar nutrient contents, enabling us to effectively test the influence of site quality on litter decomposition. For the standard litter material, we characterized the lignin content, total organic carbon (TOC), and N and P contents (Table 2).

**Table 2.** Lignin and nutrient contents in each standard litter material (mean ± standard deviation, *N* = 40).

| Plant Material | Lignin (%) | Total Organic Carbon (%) | *N* (%) | P (mg kg⁻¹) |
|---|---|---|---|---|
| *Araucaria angustifolia* | 58.6 ± 3.4 | 43.6 ± 1.4 | 0.65 ± 0.04 | 670.0 ± 45.8 |
| *Mimosa scabrella* | 34.3 ± 1.4 | 20.9 ± 1.2 | 3.95 ± 0.23 | 134.8 ± 14.7 |

### 2.3. Field Conditions, Litter Collection, and Soil Sampling

The experimental study was conducted in monospecific *A. angustifolia* stands on permanent plots at the Florestal Gateados Enterprise, Campo Belo do Sul, Santa Catarina, Southern Brazil (Figure 1).

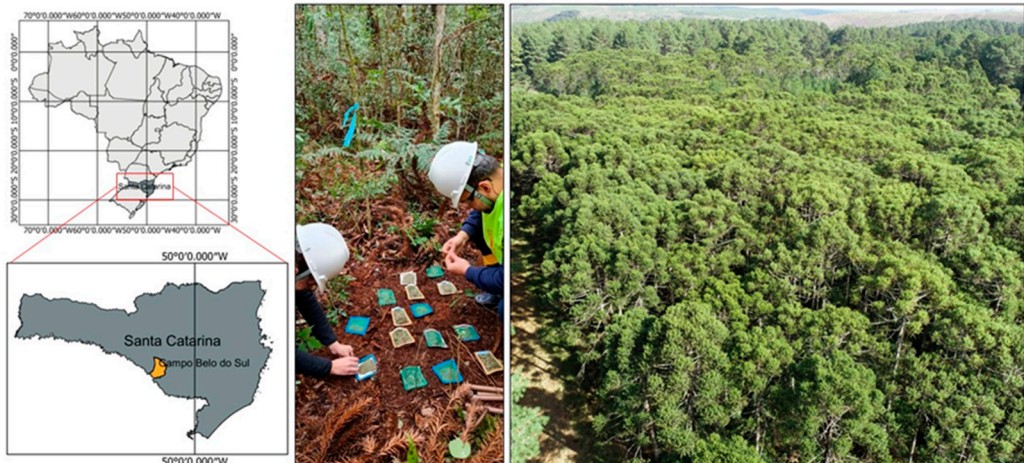

**Figure 1.** Location of our field experiment in the highlands of Santa Catarina, southern Brazil, to assess the influence of site quality levels on soil organism, litter, and soil compartments. Litterbags with 1 mm² mesh were used to determine the litter decomposition rate of the standard litter residues. Plots were monospecific and even-aged *Araucaria angustifolia* stands; an average site quality with 30-year-old trees is shown.

To determine litter deposition, litter material was collected following a 30-day schedule from August 2021 to September 2022. We used three metallic squares (1 × 1 m) per plot on the soil surface [14], and the sampling points were randomly selected before sampling by using digital map and geographic coordinates. The litter material inside the metallic square was collected using plastic bags. Litter material was air dried for 48 h at 60 °C until a constant dry biomass. We analyzed the litter nutrient contents (C, N, and P) according to Tedesco et al. [19] and lignin content using Klason's method, as described by Morais et al. [20].

Monthly, soil samples were collected from August 2021 to September 2022 using a soil auger with 6.5 cm diameter and sampled at a 0.2 m soil depth. A total of 396 soil samples (12 months × 11 plots × 3 samples per plot) were collected for soil chemical characterization. All soil samples were air dried and passed through a 2 mm sieve, as described by Black [21]. Soil pH was measured in a suspension of soil and distilled water (1:2.5, *v:v*, soil: water suspension). The available P was extracted by Mehlich-1 and determined using colorimetry. The potassium chloride extraction method was used to determine exchangeable $Al^{3+}$, $Ca^{2+}$, and $Mg^{2+}$ [21], and Mehlich-1 was used to determine $K^+$. Soil organic matter was estimated by rapid dichromate oxidation according to the methodology described by Okalebo et al. [22]. The total nitrogen was estimated using sulfuric acid (Merck, Oakville, ON, Canada) and potassium sulphate (Merck, Oakville, ON, Canada) digestion [21]. Micronutrients were estimated according to the methodologies described by Black [21].

### 2.4. Soil Organism Assemblage

To sample soil organisms, we used six Provid-type traps per plot. We did not find any nests (i.e., ants' and termites' nests) in the plots or near our experiment. Throughout the experiment, Provid-type traps (each trap had four windows with a 4 cm × 4 cm opening) were placed using a 2-day schedule without any disruption to collect soil fauna specimens (e.g., we placed the traps 12 times following a 30-day schedule throughout the experiment), but we presented the mean in our results. In each trap, we added 200 mL

of distilled water and neutral detergent solution at a concentration of 25% with 30 mL of 70% alcohol [14]. Soil fauna specimens (macrofauna—specimens longer than 0.2 cm and mesofauna—specimens shorter than 0.2 cm) were preserved in 70% alcohol and considered for our analyses. All individuals were then sorted, counted, and classified at the family level. The community structure of soil fauna in each studied treatment was characterized by the following parameters: mean abundance (ind. trap$^{-1}$) of soil fauna taxonomic groups, fauna richness, Shannon diversity index (H'), and Simpson dominance index (C). In addition, we classified the functional group of each taxonomic unit following the criteria described by Rodriguez et al. [23].

### 2.5. Litter Decomposition Assay

We used 144 litterbags (10 cm × 10 cm) with 1 mm$^2$ mesh to determine the litter decomposition rate (k-factor, years$^{-1}$) of standard litter residues (e.g., *A. angustifolia*, and *M. scabrella*), according to the methodology described by da Silva et al. [14]. The use of litterbags with 1 mm$^2$ enabled us to assess (i) macrofauna action on litter fragmentation and (ii) microbiota action on litter decomposition. Each litterbag received 10 g of dry litter residue. Litterbags were placed in a grid scheme per plot distributed between the topsoil and litter material. Following a 30-day schedule, six litterbags per standard litter residue were collected and placed in individual paper bags. In the lab, the standard litter residues sampled in each litterbag were oven-dried at 60 °C until reaching a constant weight for 72 h, and then the samples were weighted. The k-factor (years$^{-1}$) was calculated as described by da Silva et al. [14]. In this assay, the k-factor represented the rate of litter decomposition.

### 2.6. Statistical Analyses

All variables were tested for normality and homoscedasticity using the "shapiro.test" and the "bartlett.test" functions in the "stats" and "dplyr" packages, respectively. The "Moran.I" function in the "ape" package was used to detect spatial autocorrelation among blocks, and within each block among plots. We used the analysis of variance (ANOVA) using the "aov" function (in the "stats" package) to compare the influence of the site quality treatments on soil chemical properties (hypothesis 1), soil fauna assemblage (hypothesis 2), and litter deposition and decomposition (hypothesis 3). First, an explanatory ANOVA was performed to explore all data variability among the site quality treatments, sampling period, and plots on soil chemical properties, soil fauna abundance, litter deposition, k-factor, and litter nutrient content. In this explanatory analysis, we considered site quality treatments (df = 2), sampling period (df = 11), plots (df = 10), and their interactions as sources of variation in a three-way ANOVA. Data from sampling points were nested by plots. We did not find any significant differences among sampling periods, plots, or their interaction with any of the studied variables. Consequently, we refrained from describing site effects since our analyses did not indicate spatial dependence. Moreover, repeated measures ANOVA was not employed due to the absence of significant differences across sampling periods. Thus, we conducted a one-way ANOVA, considering site quality treatments as a source of variation. Here, plots and sampling periods were used as replicates. The results of this analysis are detailed in the following subsections: "Litter Deposition and Litter Nutrient Content from the Studied Site Quality Treatments", "Influence of Araucaria's Site Quality on Soil Chemical Properties", "Soil Organism Collection in an *A. angustifolia* Field", and "K-Factor and its Correlation with Litter Nutrient Content from Low and High C:N Ratio Residues in *A. angustifolia* Fields".

Pearson's correlation was used to test the correlation between the k-factor and litter nutrient content. To evaluate the similarities among the site quality treatments due to the soil chemical properties, litter deposition, k-factor, and litter nutrient contents, a principal component analysis (PCA) was carried out using the "vegan" package. We performed nonmetric multidimensional scaling (NMDS) to analyze differences among the site quality treatments in terms of the soil fauna community using the "metaMDS" function with the Bray–Curtis matrix. Structural equation model (SEM) analysis was performed to explore

the direct effect of site quality treatments on soil fauna abundance and the indirect effects via changes in soil and litter properties [13]. The variables used in the SEM were previously selected by PCA. To construct the measurement model, we utilized confirmatory factor analysis to specify the number of variables and their associated observed indicators based on loading scores. To ensure the factorability of our data, we conducted the Kaiser–Meyer–Olkin and Bartlett's tests, assessing (i) eigenvalues; (ii) cumulative variance; and (iii) factor loadings based on the selected variables and their respective indicators. The results confirmed the suitability of our data for factor analysis. Our resulting model represents a single-level analysis, incorporating hierarchical soil, litter, and biota variables for each plot [15]. The SEM approach was performed using the "psych", "lavaan", "semTools", and "MASS" packages. The maximum likelihood estimation method was used to parameterize the model. Model fit was assessed by a comparative fit index (CFI) > 0.95, standardized root mean squared residual (SRMR) < 0.08, and root mean square error of approximation (RMSEA) < 0.06 [24]. All statistical analyses were run using R 3.4.0 software [25].

## 3. Results

### 3.1. Litter Deposition and Litter Nutrient Content from the Studied Site Quality Treatments

The litter deposition, lignin content, litter nutrient content (N, P, K, Ca, Mg, S, and C), and C:N ratio varied among the site quality levels. For litter N, S, and C contents, we did not find any significant differences between L-SQ and H-SQ. In H-SQ, we found the highest significant values of litter deposition ($8922.70 \pm 797.89$ g m$^{-2}$) and litter Ca ($19.50 \pm 1.97$ g kg$^{-1}$) and Mg ($3.60 \pm 1.26$ g kg$^{-1}$) contents. For L-SQ, we found the highest significant values of lignin content ($63.50 \pm 2.96\%$) and litter P content ($0.82 \pm 0.18$ g kg$^{-1}$), while for A-SQ, the highest significant value of the C:N ratio ($41.80 \pm 1.09$) was observed (Table 3).

**Table 3.** Mean values of litter deposition, lignin, nutrient content, and C:N ratio in each litter material within the site quality levels.

| Properties | L-SQ | A-SQ | H-SQ |
|---|---|---|---|
| Litter deposition (g m$^{-2}$) | 4416.28 (626.46) c | 6518.22 (652.34) b | 8922.70 (797.89) a |
| Lignin content (%) | 63.50 (2.96) a | 51.95 (1.19) b | 43.48 (0.61) c |
| N content (g kg$^{-1}$) | 10.46 (3.02) a | 8.96 (1.73) b | 10.53 (1.72) a |
| P content (g kg$^{-1}$) | 0.82 (0.18) a | 0.74 (0.11) b | 0.69 (0.12) c |
| K content (g kg$^{-1}$) | 0.56 (0.02) b | 0.93 (0.20) a | 1.11 (0.40) a |
| Ca content (g kg$^{-1}$) | 17.18 (3.62) b | 15.67 (5.05) b | 19.50 (1.97) a |
| Mg content (g kg$^{-1}$) | 2.00 (0.69) c | 2.48 (0.35) b | 3.60 (1.26) a |
| S content (g kg$^{-1}$) | 2.43 (0.78) a | 2.09 (0.57) b | 2.42 (0.36) a |
| C content (g kg$^{-1}$) | 384.30 (4.24) a | 369.05 (4.14) b | 383.60 (4.15) a |
| C:N ratio | 38.45 (1.52) b | 41.80 (1.09) a | 36.86 (0.98) c |

L-SQ = Low site quality; A-SQ = average site quality; and H-SQ = high site quality. Standard deviation in parenthesis. Similar letters indicate no significant differences among the site quality levels according to Bonferroni's test ($p < 0.05$).

### 3.2. Influence of Araucaria's Site Quality on Soil Chemical Properties

The one-way ANOVA results showed significant differences among the studied site quality levels on soil chemical properties. The highest values of soil organic matter ($7.36 \pm 1.21\%$), soil P content ($12.35 \pm 1.21$ g kg$^{-1}$), and exchangeable Ca ($8.74 \pm 0.28$ cmol$_c$ kg$^{-1}$) were observed in soil samples from H-SQ. For L-SQ, we observed the highest values of exchangeable Mg ($2.40 \pm 0.56$ cmol$_c$ kg$^{-1}$), S content ($20.65 \pm 1.29$ mg kg$^{-1}$), and B content ($0.61 \pm 0.04$ mg kg$^{-1}$). Next, we observed the highest values of exchangeable Al$^{3+}$ ($1.50 \pm 0.17$ cmol$_c$ kg$^{-1}$), H$^+$ + Al$^{3+}$ ($14.92 \pm 0.56$ cmol$_c$ kg$^{-1}$), and Zn content ($8.07 \pm 0.75$ mg kg$^{-1}$) in A-SQ. We did not find any significant differences in soil pH and Cu content between L-SQ and H-SQ (Table 4).

**Table 4.** Soil chemical properties from the studied site quality levels, Campo Belo do Sul and Capão Alto, Santa Catarina, Brazil.

| Properties | L-SQ | A-SQ | H-SQ |
|---|---|---|---|
| Soil pH | 5.30 (0.42) a | 4.97 (0.17) b | 5.56 (0.87) a |
| Soil organic matter (%) | 5.85 (1.48) b | 5.70 (1.22) b | 7.36 (1.21) a |
| P (mg kg$^{-1}$) | 10.90 (1.25) b | 9.72 (1.30) b | 12.35 (1.21) a |
| K (mg kg$^{-1}$) | 44.00 (1.27) b | 107.80 (5.09) a | 113.35 (4.12) a |
| Exchangeable Ca (cmol$_c$ kg$^{-1}$) | 5.60 (0.30) b | 4.60 (0.37) b | 8.74 (0.28) a |
| Exchangeable Mg (cmol$_c$ kg$^{-1}$) | 2.40 (0.56) a | 1.47 (0.49) b | 1.70 (0.62) b |
| Exchangeable Al (cmol$_c$ kg$^{-1}$) | 0.40 (0.05) b | 1.50 (0.17) a | 0.38 (0.05) b |
| H$^+$ + Al$^{3+}$ (cmol$_c$ kg$^{-1}$) | 9.50 (0.51) b | 14.92 (0.56) a | 10.74 (0.78) b |
| S (mg kg$^{-1}$) | 20.65 (1.29) a | 18.87 (0.81) b | 16.20 (0.43) c |
| B (mg kg$^{-1}$) | 0.61 (0.04) a | 0.42 (0.05) b | 0.43 (0.03) b |
| Cu (mg kg$^{-1}$) | 4.05 (0.12) a | 2.57 (0.13) b | 3.30 (0.16) a |
| Mn (mg kg$^{-1}$) | 50.60 (1.32) b | 56.97 (0.42) a | 57.68 (1.23) a |
| Zn (mg kg$^{-1}$) | 2.55 (0.63) c | 8.07 (0.75) a | 4.08 (0.25) b |

L-SQ = low site quality; A-SQ = average site quality; and H-SQ = high site quality. Standard deviation in parenthesis. Similar letters indicate no significant differences among the site quality levels according to Bonferroni's test ($p < 0.05$).

### 3.3. Soil Organism Collection in A. angustifolia Plantations

We identified 18 families within the soil biota community. We found the highest abundance of Araneidade in L-SQ, while H-SQ showed the lowest abundance of such organisms. For Acaridae and Scarabidae, they were not found in L-SQ (Acaridae), and in both L-SQ and A-SQ (Scarabidae). In A-SQ, we found the highest significant values for Isotomidae, Nitidulidae, Neuroptera, and Halictophagidae. Finally, for H-SQ, we found the highest values for Filistatidae, Acaridae, Carabidae, Scarabidae, Staphylinidae, Scutigeridae, Julidae, Formicidae, Larvae, Forficulidae, Mantidae, and Pseudoscorpiones. Richness showed the highest values in H-SQ and the lowest values in L-SQ. The Shannon's diversity showed the lowest value in L-SQ, while there were no differences between A-SQ and H-SQ (Table 5).

**Table 5.** Soil organism abundance (ind. trap$^{-1}$, mean ± standard deviation, $N = 99$) and ecological indices among the studied site quality levels.

| Taxonomic Group | L-SQ | A-SQ | H-SQ | *F*-Value |
|---|---|---|---|---|
| Araneae | | | | |
| Araneidae | 15.0 (1.7) a | 11.2 (2.1) b | 5.2 (1.0) c | 7.83 ** |
| Filistatidae | 0.0 (0.0) b | 15.0 (2.1) a | 15.8 (3.8) a | 9.85 *** |
| Acari | | | | |
| Acaridae | 0.0 (0.0) c | 18.7 (3.1) b | 55.0 (4.7) a | 8.63 ** |
| Collembola | | | | |
| Isotomidae | 16.5 (0.7) b | 24.0 (1.4) a | 16.0 (4.5) b | 7.08 * |
| Paronellidae | 0.0 (0.0) c | 10.5 (1.2) b | 21.4 (3.6) a | 8.63 ** |
| Coleoptera | | | | |
| Carabidae | 0.0 (0.0) c | 12.0 (2.9) b | 16.8 (3.2) a | 6.86 * |
| Scarabidae | 0.0 (0.0) b | 0.0 (0.0) b | 28.6 (3.2) a | 8.91 ** |
| Staphylinidae | 0.0 (0.0) b | 0.0 (0.0) b | 13.8 (4.5) a | 8.96 ** |
| Nitidulidae | 10.5 (0.7) b | 18.0 (2.1) a | 12.0 (3.3) b | 6.62 * |
| Chilopoda | | | | |
| Scutigeridae | 0.0 (0.0) b | 0.0 (0.0) b | 3.4 (1.1) a | 8.96 ** |
| Diplopoda | | | | |
| Julidae | 0.0 (0.0) b | 0.0 (0.0) b | 2.0 (0.7) a | 9.11 ** |

**Table 5.** *Cont.*

| Taxonomic Group | L-SQ | A-SQ | H-SQ | *F*-Value |
|---|---|---|---|---|
| Hymenoptera | | | | |
| Formicidae | 12.5 (0.7) c | 50.2 (2.9) b | 64.0 (8.3) a | 8.63 ** |
| Lepidoptera | | | | |
| Larvae | 0.0 (0.0) b | 0.0 (0.0) b | 1.8 (0.8) a | 9.01 ** |
| Dermaptera | | | | |
| Forficulidae | 0.0 (0.0) b | 1.7 (0.9) a | 1.8 (0.8) a | 9.86 ** |
| Mantodea | | | | |
| Mantidae | 0.0 (0.0) b | 0.0 (0.0) b | 4.4 (1.0) a | 8.91 ** |
| Neuroptera | | | | |
| Myrmeleontidae | 0.0 (0.0) b | 2.0 (0.1) a | 1.4 (0.2) b | 9.26 ** |
| Pseudoscorpiones | | | | |
| Lechytiidae | 0.0 (0.0) b | 0.0 (0.0) b | 3.0 (0.8) a | 9.11 ** |
| Thysanoptera | | | | |
| Halictophagidae | 1.5 (0.7) b | 2.7 (0.9) a | 1.0 (0.2) b | 8.13 ** |
| Ecological indices | | | | |
| Biota richness | 5.0 (1.0) c | 11.0 (1.5) b | 18.0 (2.0) a | 19.25 *** |
| Shannon's diversity | 1.44 (0.25) c | 2.19 (0.51) a | 2.31 (0.33) a | 10.12 *** |

L-SQ = low site quality; A-SQ = average site quality; and H-SQ = high site quality. Standard deviation in parenthesis. Similar letters indicate no significant differences among the site quality levels according to Bonferroni's test ($p < 0.05$). *, **, and *** represent $p < 0.05$, 0.01, and 0.001, respectively.

### 3.4. Multivariate Analysis

According to the NMDS analysis, the soil organism abundance was significantly dissimilar among the study sites. The ordination of the soil organisms (Formicidae, Isotomidae, Carabidae, Acaridae, Scarabaeidae, Julidae, Scutigeridae, Staphylindae, and Mantidae) and ecological indices (biota richness and Shannon's diversity) at each studied site quality level had a stress value of 0.08. Formicidae was highly correlated with L-SQ stands, whereas Isotermidae, Carabidae, and Acaridae were highly correlated with A-SQ. Scarabaeidae, Julidae, Scutigeridae, Staphylindae, Mantidae, and ecological indices (biota richness and Shannon's diversity) were highly correlated with H-SQ stands. Together, the soil organism abundance and ecological indices explained 91.2% of the dataset variance (Figure 2).

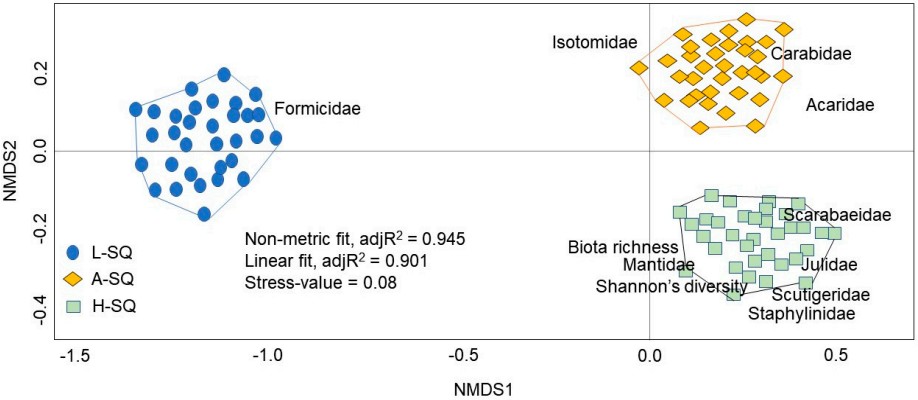

**Figure 2.** Site quality dissimilarities based on soil biota abundance plotted as nonmetric multidimensional scaling (NMDS) of the dataset from all studied site quality levels (L-SQ = low site quality; A-SQ = average site quality; and H-SQ = high site quality). Polygons represent the studied plots by each site quality, and nonmetric fit explains 91.2% of the dataset variance.

### 3.5. k-Factor Low and High C:N Ratio Residues in A. angustifolia Plantations

The two-way ANOVA results showed significant differences among the studied site quality levels and litter residues (high vs. low C:N ratio) on the k-factor ($p < 0.05$). By comparing the k-factor between litter residues with high and low C:N ratios, the residue with a low C:N ratio (*M. scabrella*) was significantly higher at all studied site quality levels. However, by comparing the k-factor among the studied site quality levels, L-SQ was useful in showing the highest values of k-factor on litterbags that received litter residues with a high C:N ratio. When residues with a low C:N ratio were added to the studied sites, the highest values were found in H-SQ (Figure 3).

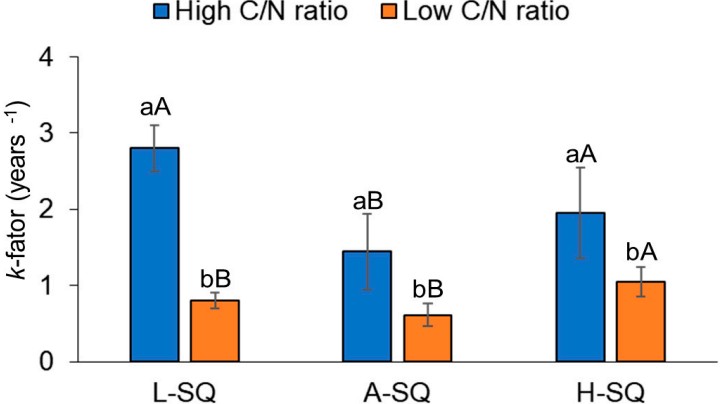

**Figure 3.** Litter decomposition (k-factor) determined by comparing decomposition rates in litter bags with different litter materials (high C:N ratio vs. low C:N ratio). Small letters compare the studied litter residues (high vs. low C:N ratio), while capital letters compare the site quality levels by the Bonferroni test at $p < 0.05$.

The structural equation model (SEM) revealed that among the *A. angustifolia* plantations, reductions in site quality were associated with negative impacts on the soil, litter, and biota compartments. Specifically, in the soil compartment, we observed increases in soil pH and $Mg^{2+}$. In the litter compartment, there were increases in k-factor, lignin, and C:N ratio. Additionally, in the biota compartment, we noted increases in the abundance of Araneidae, Carabidae, and Mantodea. Furthermore, across litter, soil, and biota compartments, it was evident that all soil and litter properties within plots subject to site quality testing exhibited decreases in exchangeable Ca content, while the abundance of Acaridae was negatively influenced (Figure 4).

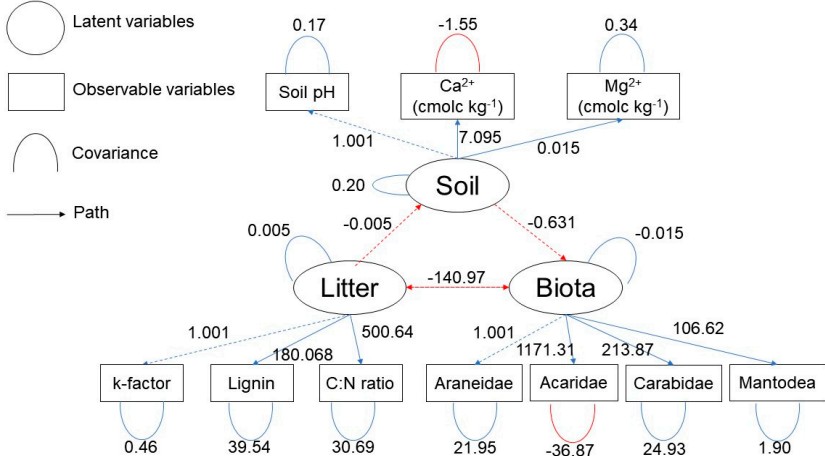

**Figure 4.** Structural equation model (SEM) indicating the effect paths of soil, litter, and biodiversity indices as influenced by low site quality level ($\chi^2 = 152.29$, $p > 0.05$, RMSEA = 0.027, CFI = 0.92). Soil

was presented by soil pH and exchangeable Ca and Mg; litter was presented by k-factor, lignin, and C:N ratio; and biota was presented by Acaridae, Araneidae, Carabidae, and Mantodea. Arrows represent significant effects at $p < 0.01$ (dashed arrows) and $p < 0.001$ (full arrows). Blue (positive) and red (negative) values represent estimated values that indicate the proportion of response variation explained by relationships with other variables. Values associated with arrows indicate standardized path coefficients.

## 4. Discussion

Our findings underscore the significant influence of contrasting site qualities within *A. angustifolia* plantations on various ecosystem components, including litter traits (deposition, nutrient content, lignin content, and k-factor), soil chemical properties, and soil biota communities. This observation aligns with previous studies demonstrating the profound impacts of tree species composition and diversity on ecosystem processes [26,27]. Specifically, our study aimed to elucidate the mechanisms underlying the effects of site quality in *A. angustifolia* plantations on litter traits, soil chemical properties, and soil biota composition. Through a long-term field experiment conducted across three site quality levels, characterized by the monodominance of *A. angustifolia*, we sought to understand the potential for site quality variations to induce negative changes within the soil ecosystem [27].

Our results revealed notable alterations in litter traits, such as higher deposition in the H-SQ when compared with the L-SQ, and shifts in nutrient content, consistent with previous research highlighting the role of plant species composition in shaping litter quality [26,28]. Furthermore, changes in soil chemical properties, including alterations in pH and nutrient availability, were evident across different site quality levels, indicating potential impacts on soil fertility and ecosystem functioning [29]. Moreover, shifts in soil biota communities, as indicated by changes in abundance, diversity (e.g., Shannon's diversity index), and richness, underscored the intricate relationships between plant species composition and belowground biodiversity [26,27].

By integrating long-term field data with insights from the existing literature, our study provides valuable insights into the complex dynamics of ecosystem functioning within *A. angustifolia* plantations and highlights the importance of considering site quality variations in ecosystem management and conservation strategies. In L-SQ stands, we found significant lower values of litter deposition, litter K, Ca, and Mg contents, and C:N ratio compared to the other sites [30,31], which in turn reduced the soil pH, soil organic matter, soil nutrient content (P, K, Ca, Mg, Mn, and Zn) [32], and soil biota abundance, richness, and diversity [33]. These results supported our first hypothesis that low site quality promotes negative effects on soil fertility with low values of soil organic matter, and nutrient content [8,26].

Other studies have reported that L-SQ in a forest ecosystem can influence litter deposition, litter nutrient contents, and the C/N ratio through various mechanisms: (i) Reduced plant productivity: Low site quality often corresponds to poor soil conditions, such as nutrient deficiency or compacted soils, which can limit plant productivity. As a result, there may be less plant biomass production, leading to lower litter deposition rates in these areas [34]; (ii) Altered litter quality: Plants growing in low-quality sites may allocate fewer resources to litter production or produce litter with different chemical compositions compared to those in high-quality sites. This can result in differences in litter nutrient contents, with litter from low-quality sites potentially containing lower concentrations of nutrients such as nitrogen, phosphorus, and potassium [35]; (iii) Slower decomposition rates: Litter decomposition rates are influenced by litter quality, with high-quality litter decomposing more rapidly than low-quality litter. Litter from low-quality sites may have higher lignin content and higher C/N ratios, which can slow down decomposition rates as microorganisms require more time and energy to break down these complex compounds [36]; (iv) Shifts in microbial communities: Low site quality can affect soil microbial communities, which play a crucial role in litter decomposition and nutrient cycling. Changes in microbial diversity and activity in low-quality sites may lead to slower decomposition rates and alterations in litter nutrient dynamics [37]; (v) Nutrient limitation: Low site quality often corresponds to

nutrient-poor soils, which can limit nutrient availability for litter decomposition processes. This nutrient limitation can further slow decomposition rates down and influence litter nutrient contents and the C/N ratio [29].

We must consider that changes in litter factors and soil properties may disrupt the soil biota community composition by altering habitat provision (e.g., when reducing litter deposition and litter lability), soil organic matter, and resource availability (litter and soil nutrients) [13]. The habitat provision and nutrient dependency hypothesis posits that soil biota rely on the supply of nutrient-rich compounds derived from litter decomposition to establish and maintain a diverse soil food web, characterized by high trophic levels of soil organisms. This hypothesis emphasizes the critical role of organic matter decomposition in sustaining soil biodiversity and ecosystem functioning [38]. There is scientific evidence demonstrating the importance of site quality and the litter compartment as key factors for nutrient cycling in some worldwide subtropical ecosystems [39,40]. Low site quality in subtropical ecosystems can cause rapid land degradation by decreasing soil organic matter, increasing the content of low labile residues, and nutrient loss by run-off, which, over time, reduces soil organic matter decomposition and nutrient cycling [41].

In the context of *A. angustifolia* plantations, the soil biota is important for promoting ecosystem services, such as litter transformation (by incorporating litter on soil profile), soil organic matter decomposition (by improving nutrient cycling), and biological control (through the activity of macro- and microregulators, also described as predators, such as Arachnids and Myriapods, and some insects, such as the Mantodea order) [42]. Soil biota also helps the soil ecosystem and plant community by providing habitats, organic matter, and nutrients [43]. The results of this study revealed that there were significant differences among the site quality levels (especially L-SQ) in soil biota abundance, richness, and Shannon's diversity [44]. Low site quality can significantly alter the soil biota composition by reducing the abundance of predators (Filistatidae, Acaridae, Carabidae, Scutigeridae, Forficulidae, Mantidae, and Pseudoscorpiones) and improving the abundance of ants (Formicidae) [45]. However, average and high site quality stands have promoted a wide range of soil biota groups through their litter and soil compartment characteristics [6]. The groups of Arachnids (Acaridae, Araneidae, and Filistatidae—classified as predators), Insects (Formicidae, Isotermidae, Carabidae, Scarabaeidae, Staphylindae, and Mantidae, which are classified as ecosystem engineers, microregulators, litter transformers, and predators), and Myriapods (Scutigeridae—predators and Julidae—litter transformers) were the most affected by reducing the site quality [46]. These changes in the soil biota community structure promoted negative chances for the soil ecosystem [47]. *Araucaria angustifolia* stands present a wide dissimilarity among the site quality levels [3], and when we decided to start our field study, we expected to find a high impact of site quality on all soil biota functional groups. However, we only observed a significant impact on predator abundance after performing SEM analysis [48].

Our hypothesis that low site quality promotes negative effects on the soil biota community composition was supported [49]. The soil biota community in this condition was characterized by a significant abundance of ants (Formicidae—ecosystem engineers). The decrease in litter deposition and litter quality (nutrient content) decreased the soil biota abundance and diversity, since litter deposited on the soil surface is considered the main source of habitat and the first stage of soil organic matter with an initial degree of decomposition [50]. For L-SQ, the lowest levels of soil biota diversity corresponded to the lowest values of litter deposition, as described by Souza et al. [13]. It also agrees with previous works carried out by Cappelli et al. [51], Lozano et al. [52], and Wang et al. [53], who reported positive correlations with (i) soil biota diversity and litter deposition; (ii) soil biota abundance and soil organic matter content; and (iii) litter quality and soil biota richness.

For litter deposition, litter traits (lignin and litter macronutrient content), and C:N ratio, H-SQ stands showed higher values of litter deposition and litter N, K, Ca, Mg, S, and C contents than the other studied site quality levels. These results support our third hypothesis that high litter deposition and litter quality may improve the soil ecosystem

through the k-factor (decomposability), nutrient cycling, and the abundance of a diverse soil biota community [5]. In fact, litter acts in two dissimilar pathways: (i) creating a habitat for soil organisms, as described by da Silva et al. [14]; and (ii) providing plants and organisms with essential nutrients [54]. Both pathways are supported by our results for litter traits and soil biota abundance in H-SQ. Our experiment was designed to directly determine whether litter properties affect soil biota biodiversity. In this context, we must consider that the highest diversity and litter properties were positively correlated by analyzing our NMDS and SEM results. According to Deng et al. [4], litter residues that present a high macronutrient content and a low lignin content are the preferred sources for r- and k-strategists that provide litter biochemical decomposition [55]. Thus, we must consider that litter residues deposited aboveground with high contents of N, K, Ca, Mg, and S may be positively correlated with high N, K, Ca, Mg, and S cycling into the soil ecosystem [56]. We cannot exclude the habitat quality and nutrient hypotheses described by Jones et al. [57], in which litter residue drives soil biota diversity in tropical and subtropical areas, which can be supported by results from the H-SQ plots. These results agree with previous studies carried out by Wang et al. [53,58], who reported that a constant input of high-quality litter may directly influence nutrient cycling and soil biota community composition by improving the abundance of some soil organisms, such as Formicidae, Isotomidae, Carabidae, Acaridae, Scarabaeidae, Julidae, Scutigeridae, Staphylindae, and Mantidae.

The three groups formed on NMDS supported our hypothesis that different site quality levels create different habitats with specific values of litter deposition, litter quality, and soil organic matter, which in turn may change the abundance of soil biota community composition. According to a study carried out by Zheng et al. [59], site quality levels and the soil biota community may create a specific pairing between them. This specific pairing can also (i) promote the dominance of some groups, such as Formicidae in L-SQ stands [60]; (ii) reduce the abundance of predators [61]; (iii) change litter dynamics, litter nutrient quality, and its decomposability, which directly affects nutrient cycling and soil biota community diversity [59]; and (iv) change soil chemical properties, which contributes to a negative tripartite relationship among soil properties, biota abundance, and litter traits as influenced by site quality for *A. angustifolia* stands [62]. This study was not designed to estimate root activity, but another study that was performed in the same experimental area provided evidence for high root activity in H-SQ stands [3]. Those authors reported that high root activity in H-SQ stands may positively influence the soil biota near the rhizosphere of *A. angustifolia* and may alter the soil reaction through root exudation and $H^+$ extrusion processes [63,64]. Interestingly, these phenomena are indirectly shown by the NMDS and SEM analyses, where dissimilarities among the site quality sites on soil biota community composition and the correlations among the soil ecosystem, abundance of some soil organisms, and litter traits were observed. Based on our results, we presume that different site quality levels can promote different habitats for root growth and specific soil food webs near the rhizosphere of *A. angustifolia* [65,66].

The soil chemical properties, especially soil acidity, and the contents of exchangeable Ca and Mg had low values in L-SQ stands due to litter traits with a high lignin content (low decomposability), high k-factor, and high C:N ratio. Both compartments combined reduced the refuge for predators, such as Araneidae, Acaridae, Carabidae, and Mantidae [4,47]. Considering some relationships among the soil biota organisms, soil ecosystem, and litter traits, we found negative effects on these three compartments, as influenced by site quality reduction [67]. In L-SQ stands, we found a significant decrease in the abundance of predators from Araneidae, Acaridae, Carabidae, and Mantidae, which may be related to active biological control. These results agree with the studies carried out by Gomez et al. [24] and Queiroz et al. [68], who reported a low abundance of predators, especially Arachnids, in soils with a low content of soil organic matter and low litter deposition. In L-SQ, we found an ecological impact described as habitat simplification. In this context, soil and litter traits may promote habitats for predators. Once the abundance of this functional

group is reduced, the abundance of other groups is directly and indirectly increased, such as the abundance of Formicidae [29].

## 5. Conclusions

L-SQ had the highest impacts from litter factors, soil chemical properties, and soil biota abundance, richness, and diversity in Cambisol soil in field conditions for *A. angustifolia* plantations. Our findings suggest that site quality decreases because of decreases in litter quantity and quality (nutrient content), soil fertility, and the abundance and diversity of soil organisms classified as predators. The results of our study highlight the fact that a fertile soil, a soil enriched in organisms, and enough litter support the forest productivity. Thus, the use of high-quality sites may exploit positive feedback among litter (by improving litter deposition, k-factor, and exchangeable Ca and Mg contents), soil properties (by improving soil organic matter, soil P content, Mn and Cu contents, and exchangeable Ca), and soil biota (by improving biota richness, diversity, and the abundance of Scarabaeidae, Julidae, Scutigeridae, Staphylindae, and Mantidae).

**Author Contributions:** Conceptualization, T.S. and M.D.J.; methodology, T.S., D.J.A., L.J.R.d.S. and M.D.J.; software, T.S., D.S.B. and D.J.A.; formal analysis, T.S., D.S.B. and M.D.J.; investigation, T.S., D.J.A., L.J.R.d.S., G.d.S.N. and M.D.J.; resources, T.S. and M.D.J.; data curation, T.S.; writing—original draft preparation, T.S., D.S.B., L.J.R.d.S. and G.d.S.N.; writing—review and editing, T.S. and M.D.J.; visualization, T.S.; supervision, T.S.; project administration, T.S. and M.D.J. All authors have read and agreed to the published version of the manuscript.

**Funding:** This work was partly funded by the Florestal Gateados, Brazil, and FAPESQ-PB, Brazil. D.S.B was funded by the National Council for Scientific and Technological Development (CNPq, Brasília, DF, Brazil: Grant no. PQ304214/2022-1). T.S. was funded by the Paraíba State Research Foundation (FAPESQ), Brazil, Grant #09-2023.

**Data Availability Statement:** Data will be made available on a reasonable request.

**Acknowledgments:** The authors are grateful to Florestal Gateados for providing all support and data about the history and soil management of the experimental area. We thank the GEBIOS (Soil Biology Research Group) for practical support. We thank the Postgraduate Program of Agroecology of the Federal University of Paraiba and the Postgraduate Program of Forest Engineering of the University of the State of Santa Catarina for facilitating the postdoctorate studies of the first author. Tancredo Souza is supported by a research fellowship from FAPESQ-PB, Brazil.

**Conflicts of Interest:** The authors have approved the manuscript and agree with the submission of the final version of the manuscript. We have read and have abided by the statement of ethical standards for manuscripts submitted as a scientific manuscript.

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
