# Peer review of "Site Quality for Araucaria angustifolia Plantations with Subtropical Cambisol Is Driven by Soil Organism Assemblage and the Litter and Soil Compartments"

_forests, doi:10.3390/f15030510_

Round 1

Reviewer 1 Report

Comments and Suggestions for Authors

The authors of the present manuscript dealt to find the relationships between A. angustifolia height and the litter floor, soil and soil biota. The research performed by the authors is interesting but the manuscript needs a substantial revision.

Ls 18-30: the abstract must be edited. You did not test the effect of soil quality, but you explored how soil change in plantations with different productivity. The reported results do not give any useful information, they are to generic.

Ls 34-55: this part needs to be re-written. There is no a clear flow of concepts, it is somhow confuse. For example it is not clear if litter is included within soil or it is another compartment. In my opinion, the authors should clearly define the soil quality, then they can introduce the concepts about soil compartments whose interaction influence the soil quality.

Ls 36-38: delete this part or move it to the end of introduction sectionz.

L 42: "....soil as influenced by soil quality".....it sounds like "soil is influenced by soil itself".

Ls 66-71: what do we know about soil in subtropical ecosystems? In order to highlight the knowledge gap, you should report the state of art about tropical soils. After that, you can use the Aracuria angustifolia plantation as an example that you took into account for increase the knowledge in such ecosystems.

Ls 80-86: this part can be shortened. It includes some repetitions.

Ls 86-91: very long sentence. You can split it into at least 2 hypotheses

L 92: the Meterials and Methods section is not well organized. Please edit it

Ls 95-96: "well-defined site quality levels", what do you mean? At L 85 you reported that it is referred to the yield. Therefore, how much yield is for low, average and high quality sites?

Table 1: although the clays have the major role on nutrient retention and dynamics, the processes related to them are influenced also by the amount of sand and silt

Table 1: number of plots and number of soil samples, what are you referring to? You had 11 plots per site. What do you mean for 99?

Ls 108-109: move this part to the introduction section in order to define the site quality

Ls 111-124: the preparation of such standard litter material does not make sense. Since your aim was to compare 3 ecosystems with different productivity, to compare the data obtained from them with those got from an artificial ecosystem (cultivation in pots in greenhouse) has no sense. It is like to compare apple and pear. Maybe it could be useful if you collected the fresh leaves from the plants of the studied ecosystems. Remove this part from here, results and discussion sections.

Ls 128-135: this part should be moved to the subsection 2.1

Ls 142-143: in total, how many litter samples did you collect? did you collect one litter sample per month per plot (L 97)? Further, in July 2021 did you remove all the litter layer before to perform in August 2021 the first sampling? Otherwise you cannot consider within your study such data. Did you georeferenced the sampling points? Otherwise, how did you measure the monthly litter deposition?

L 149: here 72 soil samples, within Table 1 594 soil samples. Be clearer. For example if you sampled once per month within the eleven plots of each site quality (12 X 11 X 3) you have to obtain 396 soil samples in total.

Ls 180-192: similar to a previous comment. This part has no sense. The authors should use the fresh litter of the studied sites. Although the litterbag experiment was well performed, the results are biased by the plant material which could not have the same properties of the plant material of studied sites. In addition, since your study sites were composed of A. angustifolia, it is not clear why you performed the experiment with residues of other plant species.

Ls 199-200: since for each plot you measured the parameters multiple times in one year, you should use one way repeated measures ANOVA.

L 212: on which base was such selection?

L 220-233: it is somehow a repetition (but with more details) of a previous one. In my opinion the ANOVA should be re-performed because the plots are your replicates and you performed the sampling multiple times per year.

Table 3: delete the column F-value, in addition the abbreviation L-SQ.....must be explained with full name within the caption or within the footnote. For the means, use 3 significant figures both for the means and the standard deviations. Avoid to use the bold type for the values within the tables.

L 256: "...pH and Cu...."

Ls 266-75: this part needs to be rewritten. For example, for Araneidae (Ls 266-267) you should report that the highest values were observed in L-SQ, while H-SQ showed the lowest abundance of such organisms. Cases like Acaridae and Scarabidae, it is important to mention that they were not found in L-SQ (Acaridae) and in both L-SQ and A-SQ (Scarabidae). At Ls 271-272 you should report that the biota richness is highest in H-SQ and lowest in L-SQ. The Shannon's diversity showed the lowest value in L-SQ, while there were no differences between A-SQ and H-SQ.

Ls 281-289: I'm not sure that the NMDS was performed correctly. I think that it does not work with missing values unless your data was 0. In this case, in Table 5 replace the dash with 0.

L 297: see a previous comment

Ls 312-319: did you perform the SEM only for the L-SQ? In materials and methods you reported that "SEM analysis was performed to explore the direct effect of soil quality treatments on soil fauna abundance and the indirect effects via changes in soil and litter properties"

Ls 330-450: the discussion section needs to be deeply revised. Some parts are merely a list of results without any explanation of the mechanisms related to them (e.g., Ls 338-340, Ls 361-375). Some parts sounds like a didactical part (Ls 341- 352). Some sentences have no sense (e.g., Ls 432-433), the authors reported the findings of previous studies without to provide a clear helpfulness of them for their own findings (e.g., Ls 414-421). In addition, the authors neglected the plant height (plant yield), namely the authors did not cite any paper that relate soil quality with plant yield.

Ls 331-337: it is not necessary to write again such information. Delete this part.

L 338: you did not find a decrease of litter deposition but you found a lower litter deposition compared to the other sites.

Ls 452-464: although the plant productivity influences the litter, soil, and biota properties, the soil environment (litter, soil, soil biota) affect the growth of the plants. Hence, the conclusion should highlight the fact that a ferile soil, a soil enriched of organisms and a sufficient amount of litter support the forest productivity. Therefore it is important to promote such features. While, in the present form, the conclusion seems to highlight that we have to take care only to the H-SQ sites because they are the most productive.

Comments on the Quality of English Language

English of the text needs a minor revision

Author Response

The authors of the present manuscript dealt to find the relationships between A. angustifolia height and the litter floor, soil and soil biota. The research performed by the authors is interesting, but the manuscript needs a substantial revision.

Thank you for your feedback. We carefully revise the entire manuscript following your suggestions.

Ls 18-30: the abstract must be edited. You did not test the effect of soil quality, but you explored how soil change in plantations with different productivity. The reported results do not give any useful information, they are to generic.

Agreed. We have provided more information and numbers in the abstract. See L18-19; 22-30; and 35-36.

Ls 34-55: this part needs to be re-written. There is no a clear flow of concepts, it is somehow confuse. For example it is not clear if litter is included within soil or it is another compartment. In my opinion, the authors should clearly define the soil quality, then they can introduce the concepts about soil compartments whose interaction influence the soil quality.

Agreed. We have started the introduction with concept of site quality, and then we have introduced the concepts about soil compartments following your suggestions. See L40-48.

Ls 36-38: delete this part or move it to the end of introduction section.

We have deleted this statement accordingly.

L 42: "....soil as influenced by soil quality".....it sounds like "soil is influenced by soil itself".

We have adjusted it to be clear. See L50-53.

Ls 66-71: what do we know about soil in subtropical ecosystems? In order to highlight the knowledge gap, you should report the state of art about tropical soils. After that, you can use the Aracuria angustifolia plantation as an example that you took into account for increase the knowledge in such ecosystems.

We got the reviewer’s point-of-view. Thus, we have improved what do we know about subtropical ecosystems, and we have added a scientific statement about that. See L80-87.

Ls 80-86: this part can be shortened. It includes some repetitions.

Agreed. We have adjusted it accordingly. See L76-80.

Ls 86-91: very long sentence. You can split it into at least 2 hypotheses

Agreed. We have adjusted it accordingly. See L96-104.

L 92: the Meterials and Methods section is not well organized. Please edit it

Thank you. We have organized this section following your comments.

Ls 95-96: "well-defined site quality levels", what do you mean? At L 85 you reported that it is referred to the yield. Therefore, how much yield is for low, average and high quality sites?

We deleted the term 'well-defined' for clarity and conciseness. Additionally, we clarified the meaning of the site quality degrees based on tree height. See L107-111.

Table 1: although the clays have the major role on nutrient retention and dynamics, the processes related to them are influenced also by the amount of sand and silt

Agreed. Thus, we have added the silt and sand contents in the table 1.

Table 1: number of plots and number of soil samples, what are you referring to? You had 11 plots per site. What do you mean for 99?

We apologize for our mistake. These numbers are presented incorrectly. There is any sense these two columns, thus we deleted them. We presented silt and sand contents instead number of plots and number of samples.

Ls 108-109: move this part to the introduction section in order to define the site quality

We defined site quality accordingly. See L40-43.

Ls 111-124: the preparation of such standard litter material does not make sense. Since your aim was to compare 3 ecosystems with different productivity, to compare the data obtained from them with those got from an artificial ecosystem (cultivation in pots in greenhouse) has no sense. It is like to compare apple and pear. Maybe it could be useful if you collected the fresh leaves from the plants of the studied ecosystems. Remove this part from here, results and discussion sections.

The reviewer raises an interesting point in this subsection. We respect her/his opinion, but we did not agree with it. Let us justify because we did not agree with this opinion because our method led us to a correct interpretation of litter decomposition without external sources of variations. Our litter preparation was meticulous, ensuring several important aspects: (i) maintaining QA/QC quality by using litter material of the same age for two materials by considering their C/N ratio; (ii) ensuring the litter material was free of pathogens and decomposers by collecting it under aseptic conditions in a greenhouse, thus preventing variations in decomposition stages that could significantly impact the litter decomposition assay; and (iii) maintaining similar nutrient contents, enabling us to effectively test the influence of site quality on litter decomposition.

Collecting fresh litter under field conditions would have made it challenging to ensure these three points, even with careful collection practices. How could we guarantee that the fresh leaves of Araucaria collected in the field had the same age and therefore the same leaf traits? Additionally, how confident could we be that the fresh litter collected under field conditions was free of soil particles, fungi spores, and other opportunistic pathogens, thus serving as a standard material? Moreover, comparing the results of decomposition using litter with different ages, nutrient contents, and decomposition degrees would be akin to comparing apples and pears.

We recognize that we may not have sufficiently clarified these points in the manuscript and have made adjustments accordingly. See L134-141.

Ls 128-135: this part should be moved to the subsection 2.1

Agreed. See L117-121.

Ls 142-143: in total, how many litter samples did you collect? did you collect one litter sample per month per plot (L 97)? Further, in July 2021 did you remove all the litter layer before to perform in August 2021 the first sampling? Otherwise you cannot consider within your study such data. Did you georeferenced the sampling points? Otherwise, how did you measure the monthly litter deposition?

We collected litter once per month. We used three metallic square (1 × 1 m) per plot on the soil surface, and the sampling points were randomly selected before sampling by using digital map and geographic coordinates. We have clarified it. See L157-160.

L 149: here 72 soil samples, within Table 1 594 soil samples. Be clearer. For example if you sampled once per month within the eleven plots of each site quality (12 X 11 X 3) you have to obtain 396 soil samples in total.

We apologize for our mistake. You are correct. We have collected a total of 396 soil samples. We have adjusted it accordingly. See 166-167.

Ls 180-192: similar to a previous comment. This part has no sense. The authors should use the fresh litter of the studied sites. Although the litterbag experiment was well performed, the results are biased by the plant material which could not have the same properties of the plant material of studied sites. In addition, since your study sites were composed of A. angustifolia, it is not clear why you performed the experiment with residues of other plant species.

We have given explanation about the use of standard litter instead fresh litter. See L134-141.

Ls 199-200: since for each plot you measured the parameters multiple times in one year, you should use one way repeated measures ANOVA.

We did that, but we did not find significant differences among the sampling periods. We have given an explanation in the manuscript. See L224-229.

L 212: on which base was such selection?

Based on eigenvalues, cumulative variance, and factor loadings. See L246-251.

L 220-233: it is somehow a repetition (but with more details) of a previous one. In my opinion the ANOVA should be re-performed because the plots are your replicates and you performed the sampling multiple times per year.

Plots and sampling periods were used as replicates. See L224-230.

Table 3: delete the column F-value, in addition the abbreviation L-SQ.....must be explained with full name within the caption or within the footnote. For the means, use 3 significant figures both for the means and the standard deviations. Avoid to use the bold type for the values within the tables.

Agreed. See the current table 3.

L 256: "...pH and Cu...."

Agreed. See L283.

Ls 266-75: this part needs to be rewritten. For example, for Araneidae (Ls 266-267) you should report that the highest values were observed in L-SQ, while H-SQ showed the lowest abundance of such organisms. Cases like Acaridae and Scarabidae, it is important to mention that they were not found in L-SQ (Acaridae) and in both L-SQ and A-SQ (Scarabidae). At Ls 271-272 you should report that the biota richness is highest in H-SQ and lowest in L-SQ. The Shannon's diversity showed the lowest value in L-SQ, while there were no differences between A-SQ and H-SQ.

Thank for your suggestions. We have rewritten this paragraph by considering your comment. See L294-297; 301-303.

Ls 281-289: I'm not sure that the NMDS was performed correctly. I think that it does not work with missing values unless your data was 0. In this case, in Table 5 replace the dash with 0.

The data used was 0. We have correct table 5 accordingly.

L 297: see a previous comment

We have given explanation about the use of ANOVA. See L224-230.

Ls 312-319: did you perform the SEM only for the L-SQ? In materials and methods you reported that "SEM analysis was performed to explore the direct effect of soil quality treatments on soil fauna abundance and the indirect effects via changes in soil and litter properties"

We apologize for our mistake. The SEM was performed as described in the M&M subsection. We have corrected it accordingly. See L344-352.

Ls 330-450: the discussion section needs to be deeply revised. Some parts are merely a list of results without any explanation of the mechanisms related to them (e.g., Ls 338-340, Ls 361-375). Some parts sounds like a didactical part (Ls 341- 352). Some sentences have no sense (e.g., Ls 432-433), the authors reported the findings of previous studies without to provide a clear helpfulness of them for their own findings (e.g., Ls 414-421). In addition, the authors neglected the plant height (plant yield), namely the authors did not cite any paper that relate soil quality with plant yield.

Thank for your feedback and for indicating each fragile point in the discussion. We have rewritten it accordingly to give a deeper revision about our findings. See L364-373; 375-383; 392-412; and 416-420.

Ls 331-337: it is not necessary to write again such information. Delete this part.

Agreed. It was deleted.

L 338: you did not find a decrease of litter deposition but you found a lower litter deposition compared to the other sites.

True. See L387-389.

Ls 452-464: although the plant productivity influences the litter, soil, and biota properties, the soil environment (litter, soil, soil biota) affect the growth of the plants. Hence, the conclusion should highlight the fact that a ferile soil, a soil enriched of organisms and a sufficient amount of litter support the forest productivity. Therefore it is important to promote such features. While, in the present form, the conclusion seems to highlight that we have to take care only to the H-SQ sites because they are the most productive.

We have adjusted it accordingly. See L525-527.

Reviewer 2 Report

Comments and Suggestions for Authors

In any region of the world, soil quality is a measure of biodiversity and plays an important role in its maintenance. The authors of this research have conducted a thorough investigation by examining various soil quality levels in plantations of Araucaria angustifolia, which may have an impact on soil organisms and the interaction between soil compartments and litter. The article presents previously unpublished data and overall research, and the findings are good. In my view, the article could be accepted for publication in this journal after some minor revisions, rectifying typographical errors, and formatting in its current form.

Some comments and recommendations are listed below:

1)    In line 128, A. angustifolia should be italicized.

2)      The authors should use the symbol for ‘°C’ as ‘ËšC’ throughout the manuscript.

3)      In line 266, the authors have mentioned, ‘We identified 27 families within the soil..,’ but in table 5, I found only 18 of them. The authors should clarify this or check for it.

4)      Format the tables and figures and their captions as per journal guidelines.

5)      The authors need to improve the quality of the English grammar in the manuscript.

6)      Authors should provide clear and high-resolution figures.

7)      There are some typographical errors throughout the manuscript; the authors should find them and correct them accordingly during the revision.

8)      The authors should thoroughly check all references formatted as per journal guidelines.

Comments on the Quality of English Language

Minor editing of English language required

Author Response

In any region of the world, soil quality is a measure of biodiversity and plays an important role in its maintenance. The authors of this research have conducted a thorough investigation by examining various soil quality levels in plantations of Araucaria angustifolia, which may have an impact on soil organisms and the interaction between soil compartments and litter. The article presents previously unpublished data and overall research, and the findings are good. In my view, the article could be accepted for publication in this journal after some minor revisions, rectifying typographical errors, and formatting in its current form.

We carefully revised all minor aspects, and we are so glad to receive your feedback. Thank you.

Some comments and recommendations are listed below:

1)    In line 128, A. angustifolia should be italicized.

Agreed. See L147.

2)      The authors should use the symbol for ‘°C’ as ‘ËšC’ throughout the manuscript.

Agreed. See L118.

3)      In line 266, the authors have mentioned, ‘We identified 27 families within the soil..,’ but in table 5, I found only 18 of them. The authors should clarify this or check for it.

Agreed. See L294.\

4)      Format the tables and figures and their captions as per journal guidelines.

We have adjusted it accordingly. We have used the provided template.

5)      The authors need to improve the quality of the English grammar in the manuscript.

All manuscript was revised by considering the English grammar.

6)      Authors should provide clear and high-resolution figures.

We have provided them accordingly.

7)      There are some typographical errors throughout the manuscript; the authors should find them and correct them accordingly during the revision.

Thank you for your suggestion. We have carefully revised the entire manuscript.

8)      The authors should thoroughly check all references formatted as per journal guidelines.

We have checked all references by considering authors’ guidelines.

Reviewer 3 Report

Comments and Suggestions for Authors

This manuscript evaluated the litter and soil compartments and the soil organism community composition associated with three degrees of site quality (low-, average-, and high-quality sites), which reflect productivity levels, in the highlands of southern Brazil. The manuscript should be revised before publication publishing.

Abstract

The conclusions of this study (line 28-30) do not return to the hypothesis of the manuscript.

Introduction

1.The manuscript must show the reason for why the study was carried out clearly.

2.The ack of research on A. angustifolia is unclear is unclear.

3.Why used “the soil’s main chemical properties, litter properties, and soil organism abundance for three well-defined soil quality degrees (line 82-84).”

Materials and Methods

1.Supplementary soil auger diameter (line 149).

2.Delete the following. The soil chemical characterisation included soil pH, available P, soil organic matter, total nitrogen, exchangeable cations (Ca2+, Mg2+, K+, and Al3+), H++Al3+, sulphur, and micronutrients (line 151-153).

Results

1.Line 47.P in italics. Many similar expressions follow.

2. Please highlight the main results of this manuscript.

Discussion

The discussion section has limited support for the hypothesis of this manuscript. It is recommended to highlight the important content that you want to express. For example, add subheadings to better clarify the main points of the discussion.

Author Response

This manuscript evaluated the litter and soil compartments and the soil organism community composition associated with three degrees of site quality (low-, average-, and high-quality sites), which reflect productivity levels, in the highlands of southern Brazil. The manuscript should be revised before publication publishing.

Thank you for all your suggestions and support by revising our manuscript.

Abstract: The conclusions of this study (line 28-30) do not return to the hypothesis of the manuscript.

We have revised it accordingly. See L35-36.

Introduction

1.The manuscript must show the reason for why the study was carried out clearly.

Agreed. See L40-48.

2.The ack of research on A. angustifolia is unclear is unclear.

Agreed. See L50-53.

3.Why used “the soil’s main chemical properties, litter properties, and soil organism abundance for three well-defined soil quality degrees (line 82-84).”

We have rewritten this sentence to be clear. See L99-104.

Materials and Methods

1.Supplementary soil auger diameter (line 149).

Agreed. See L166.

2.Delete the following. The soil chemical characterisation included soil pH, available P, soil organic matter, total nitrogen, exchangeable cations (Ca2+, Mg2+, K+, and Al3+), H++Al3+, sulphur, and micronutrients (line 151-153).

Agreed. It was deleted.

Results

1.Line 47.P in italics. Many similar expressions follow.

It was revised accordingly.

  1. Please highlight the main results of this manuscript.

They were highlighted accordingly. See L263-266; L277-283.

Discussion

The discussion section has limited support for the hypothesis of this manuscript. It is recommended to highlight the important content that you want to express. For example, add subheadings to better clarify the main points of the discussion.

This section was improved. See L364-383; 392-412.

Round 2

Reviewer 1 Report

Comments and Suggestions for Authors

L 19: define the soil compartments. I suggest you to write them in brackets.

L 24: here and for the other parameters, it is not clear if (in this case) the litter deposition increased by 4,416 and 8,923 g/m2 in LSQ and HSQ, respectively, or in both sites the litter deposition increased from 4,416 to 8,923. If the former, I suggest you to report the variation as percentage

L 125: the mean values of sand, silt and clay are not %, but g kg-1

L 126: remove the asterisk

 L 265: "......(8,923 ± 798)..."

L 273: Here and for the other tables. Maybe you meant "Similar letters indicates no significant differences among the site quality levels according to Bonferroni's......"

L 275: Here and for the other tables, remove this part, you do not have asterisks

Author Response

L19: define the soil compartments. I suggest you to write them in brackets.

Agreed. We have adjusted it accordingly. See L19.

L24: here and for the other parameters, it is not clear if (in this case) the litter deposition increased by 4,416 and 8,923 g/m2 in LSQ and HSQ, respectively, or in both sites the litter deposition increased from 4,416 to 8,923. If the former, I suggest you to report the variation as percentage

Agreed. We have rewritten this sentence to be clear. See L23-25.

L125: the mean values of sand, silt and clay are not %, but g kg-1

That is correct, thanks. We have adjusted it accordingly. See L118.

L126: remove the asterisk

Agreed. See L119.

 L265: "......(8,923 ± 798)..."

Agreed. See L258.

L273: Here and for the other tables. Maybe you meant "Similar letters indicates no significant differences among the site quality levels according to Bonferroni's......"

That is correct, thanks. We have adjusted it in all tables. See Table 3, 4, and 5.

L275: Here and for the other tables, remove this part, you do not have asterisks

Agreed. See Table 3 and 4.

Reviewer 3 Report

Comments and Suggestions for Authors

Abstract

1.Delete” Our aim was to present a deeper view of the litter and soil compartments and the soil organism community composition associated with three degrees of site quality (low-, average-, and high-quality sites), which reflect productivity levels, in the highlands of southern Brazil.”

2. Add how to complete the research of this manuscript in appropriate location, briefly.

Introduction

1.“Site quality is a variable or measure used in forestry to assess the productivity of a 40

forest site, particularly in terms of tree height growth (line 40-41)” This part is your results or which you cited? There is the same problem of the part of line 43-46.

2. I did not see any “*” in the Table 1.

3. Results

1. I did not see any “*” in the Table 3 and 4.

2. P of “P-value” must be in italics.

3. There are so many results, delete some results that are not relevant to the purpose and hypothesis of this study.

Discussion

1. Although many theories are listed, the discussion in each section is very unfocused and does not answer whether the results support the hypothesis or not.

References

1.      Check the format of the reference of newly added. Such as line 608-615,626-633.

Author Response

1.Delete” Our aim was to present a deeper view of the litter and soil compartments and the soil organism community composition associated with three degrees of site quality (low-, average-, and high-quality sites), which reflect productivity levels, in the highlands of southern Brazil.”

Agreed.

  1. Add how to complete the research of this manuscript in appropriate location, briefly.

Agreed. See L20-22.

Introduction

1.“Site quality is a variable or measure used in forestry to assess the productivity of a forest site, particularly in terms of tree height growth (line 40-41)” This part is your results or which you cited? There is the same problem of the part of line 43-46.

In the entire manuscript, we base our results and discussions on the premise that soil compartments are influenced by site quality. Following the suggestion of reviewer #1 in the previous revision round, we added additional information about the definition of site quality and soil compartments. Thus, in this sentence, we define what site quality is, and after that, we define what soil compartments are.

  1. I did not see any “*” in the Table 1.

We have deleted it accordingly. See L119.

  1. Results
  2. I did not see any “*” in the Table 3 and 4.

We have deleted them accordingly. See L268, 285.

  1. P of “P-value” must be in italics.

Agreed. See L268, 285, 303.

  1. There are so many results, delete some results that are not relevant to the purpose and hypothesis of this study.

We considered the reviewer's point of view; however, all presented results are relevant to our hypothesis. In Table 3, we provide detailed information about litter deposition and its nutrient content. Table 4 describes the main soil chemical properties. For biota abundance, we referred to Table 5. Finally, to provide a multivariate overview of our dataset, we included Figures 2, 3, and 4.

Considering the text in the results section, all described results showed significant differences among site qualities. Thus, we did not delete any results. As described before, we considered all of them relevant.

Discussion

  1. Although many theories are listed, the discussion in each section is very unfocused and does not answer whether the results support the hypothesis or not.

We have rewritten some sentences to clear if the results supported or not our hypotheses. See L385-386, 445-446, and 458-461.

References

  1. Check the format of the reference of newly added. Such as line 608-615,626-633.

Agreed. See References nº: 24, 26-29, 33-36, and 57.